# Validation of an HPLC Method for the Simultaneous Quantification of Metabolic Reaction Products Catalysed by CYP2E1 Enzyme Activity: Inhibitory Effect of Cytochrome P450 Enzyme CYP2E1 by Salicylic Acid in Rat Liver Microsomes

**DOI:** 10.3390/molecules25040932

**Published:** 2020-02-19

**Authors:** Hassan Salhab, Declan P. Naughton, James Barker

**Affiliations:** School of Life Sciences, Pharmacy and Chemistry, Kingston University, Kingston-upon-Thames, London KT1 2EE, UK; d.naughton@kingston.ac.uk (D.P.N.); j.barker@kingston.ac.uk (J.B.)

**Keywords:** salicylic acid, rat liver microsomes, CYP2E1 activity, cytochrome P450, mixed inhibitor

## Abstract

Inhibition of cytochrome P450 (CYP) alters the pharmacokinetic parameters of the drug and causes drug–drug interactions. Salicylic acid been used for the treatment of colorectal cancer (CRC) and chemoprevention in recent decades. Thus, the aim of this study was to examine the in vitro inhibitory effect of salicylic acid on CYP2E1 activity in rat liver microsomes (RLMs) using high-performance liquid chromatography (HPLC). High-performance liquid chromatography analysis of a CYP2E1 assay was developed on a reversed phase C_18_ column (SUPELCO 25 cm × 4.6 mm × 5 µm) at 282 nm using 60% H_2_O, 25% acetonitrile, and 15% methanol as mobile phase. The CYP2E1 assay showed a good linearity (R^2^ > 0.999), good reproducibility, intra- and inter-day precision (<15%), acceptable recovery and accuracy (80–120%), and low detection (4.972 µM and 1.997 µM) and quantitation limit values (15.068 µM and 6.052 µM), for chlorzoxazone and 6-hydroxychlorzoxazone, respectively. Salicylic acid acts as a mixed inhibitor (competitive and non-competitive inhibition), with K_i_ (inhibition constant) = 83.56 ± 2.730 µM and concentration of inhibitor causing 50% inhibition of original enzyme activity (IC_50_) exceeding 100 µM (IC_50_ = 167.12 ± 5.460 µM) for CYP2E1 enzyme activity. Salicylic acid in rats would have both low and high potential to cause toxicity and drug interactions with other drugs that are substrates for CYP2E1.

## 1. Introduction

Members of the cytochrome P450 (CP450) superfamily are known as phase 1 enzymes and play a key role in the biotransformation of a large number of endogenous (steroids, hormones, bile acids, fatty acids) and exogenous (toxic chemicals, drugs, carcinogens, environmental pollutants) compounds to a more hydrophilic form [1]. Statistical results showed that more than 90% of marketed drugs are metabolized by P450s [2]. However, with more than 10 forms identified, only the CYP1, CYP2, and CYP3 families are involved in the biotransformation of the majority of clinical drugs [2].

Because P450s play a vital role in drug metabolism, co-administration of a drug with another CYP substrate may alter their metabolism, thus causing a drug–drug interaction [2]. As a definition, drug interactions take place when a certain drug interacts with another drug [3]. These interactions can result in the changing activity of one or both drugs and lead to adverse side effects [4]. Statistical evidence has shown that nearly 20–40% of elderly people in developing countries have drug–drug interactions due to poly-therapy [5].

Understanding the properties of adverse drug reactions (ADRs) gives us a clear indication for quantification of the side effects of a prospective drug and a good knowledge of the pathogenic pathways involved during the interaction [5]. Nowadays, adverse drug reactions seem to be the main obstacle in clinical trials, slowing down the recovery of patients in hospitals [5]. 

It has been reported by Badyal and Dadhich (2001) that drug–drug interactions could be the fourth or the sixth leading cause of deaths in United States. Drug–drug interactions can be the result of inhibition and induction of P450 enzyme activity [3]. Potential drug–drug interactions appear to be the most challenging part of clinical practice, resulting in pharmacokinetic and pharmacodynamic variations of the drugs and changing their overall therapeutic response [6]. On the contrary, a clear understanding in both drug metabolism and drug interaction has resulted from allotting P450 enzymes based on their amino acid chain [7]. Many drugs have been withdrawn from U.S. market, e.g., terfenadine, astemizole, and cisapride, because the inhibition of these drugs by other drugs results in arrhythmias [8].

Relevant studies have suggested that potential drug–drug interactions (pDDIs) are more likely to occur in persons suffering from cardiovascular disease compared to other major diseases [6]. The explanation for this clinical observation can refer primarily to many factors such as age, combination therapy, and the effective therapeutic management of drugs used in cardiology [6]. In fact, mibefradil, a calcium channel blocker, was withdrawn from U.S. market since it acts as a potent inhibitor that increases the toxicity level of certain cardiovascular drugs [8]. Specific enzyme activity and metabolite formation can be predicted. 

Aspirin (acetyl salicylic acid) is a nonsteroidal, anti-inflammatory drug used for the treatment of fever, pain, and inflammation [9]. It is hydrolysed to salicylic acid in the liver, intestine, and plasma by means of the esterase enzyme [9]. Salicylic acid (Figure 1) is a natural compound obtained from wintergreen leaves and white willow trees, and possesses many beneficial pharmacological activities, such as anti-inflammatory, antioxidant, and vasodilator effects [9]. It has been reported by Danchineni (2017) that salicylic acid potentially targets tumour cells by downregulating the function of cyclin A2, B1, D3, and cyclin-dependent kinases (CDKs 1,2,4, and 6) to exert a chemopreventive effect. Recently, inhibition studies demonstrated that CYP2E1 enzyme is responsible for the hydroxylation of salicylic acid in humans [9].

However, there is a lack of information concerning the effect of salicylic acid on other CYPs. There is thus an urgent need for a detailed study on the inhibitory effect of salicylic acid on other P450s enzyme activities. Our previous in vitro inhibition study showed that salicylic acid acts as a non-competitive inhibitor for CYP2C11 enzyme activity [10]. This means that salicylic acid has a low potency to cause drug interactions with other drugs that are substrates for the CYP2C11 enzyme.

The aim of this study is to investigate the inhibitory effect of salicylic acid on the metabolism of the CYP2E1 isoform in male rat liver microsomes. In this systematic study, the in vitro inhibitory effect of salicylic acid on CYP2E1 enzyme activity was evaluated to determine the potency of salicylic acid in affecting CYP mediated phase 1 metabolism in male rat microsomes, employing chlorzoxazone as a probe substrate for the CYP2E1 enzyme in the presence of different concentrations of salicylic acid.

## 2. Results and Discussion

### 2.1. Selection of Analytical Wavelength: UV-VIS SPECTROSCOPY (CYP2E1 Assay)

UV-VIS spectrophotometry analysis for CYP2E1 assay was carried out by dissolving chlorzoxazone (200 µM), phenacetin (50 µM), salicylic acid (100 µM), and 6-hydroxychlorzoxazone (50 µM) powder in pure acetonitrile (wavelength cut-off is 210 nm) [11].

The following figure demonstrates the measurements of maximum wavelength of each compound in the CYP2E1 assay.

According to the overlain spectra (Figure 2), it is perceived that the maximum of the absorption band for the four components in CYP2E1 assay is 282 nm.

### 2.2. Method Development (CYP2E1 Assay):

Method development for CYP2E1 assay was assessed using HPLC low-pressure isocratic elution programming (60% H_2_O, 15% methanol, and 25% acetonitrile) (λ = 282 nm, flow rate = 0.7 mL/min, T = 25 °C) (see Appendix A).

### 2.3. Validation of the Analytical Chromatographic Method (CYP2E1 Assay):

#### 2.3.1. Specificity and Selectivity

Specificity was achieved by choosing the right mobile phase composition (60% H_2_O, 25% acetonitrile, and 15% methanol). It results in a good separation of the salicylic acid peak from the CYP2E1 metabolite (6-hydroxychlorzoxazone) peak at T = 25 °C using C18 (SUPELCO 25 cm × 4.6 mm, 5 µm), at a 0.7 mL/min flow rate and wavelength of λ = 282 nm. In order to determine the retention time of each compound in the CYP2E1 assay, each compound was run separately using phenacetin as an internal standard on the HPLC instrument using isocratic elution programming. 

Figure 3 shows the chromatogram of the response peaks of four compounds (salicylic acid, chlorzoxazone, 6-hydroxychlorzoxazone, and phenacetin) in the conditions mentioned above.

#### 2.3.2. Linearity and Range

Different solution concentrations of chlorzoxazone (0, 25, 50, 100, 150, 200, 300, and 400 µM) and its metabolite 6-hydroxychlorzoxazone (0, 10, 20, 40, 60, 80, and 100 µM) were injected into the HPLC instrument for the determination of the standard calibration curves. Calibration curves were assessed by plotting the mean area peak of standards and phenacetin (internal standard) (50 µM) versus the concentration of standard (chlorzoxazone and 6-hydroxychlorzoxazone). The outcomes are listed in Table 1 and show good linearities for both chlorzoxazone and 6-hydroxychlorzoxazone, with *r^2^* values of 0.9997 and 0.9994, respectively. The linear regression coefficient was within the acceptable fit (*r^2^* > 0.99) according to the International Conference on Harmonization (ICH) guidelines. The % RSD (relative standard deviation at each different concentration (% RSD < 5%)) met the ICH guidelines.

#### 2.3.3. Limit of Detection and Limit of Quantitation (LOD and LOQ)

The results, presented in Table 2, demonstrate that both chlorzoxazone and 6-hydroxychlorzoxazone have low detection and quantitation limit values, consistent with the literature [12].

#### 2.3.4. Precision

##### Intra-assay Variation of Chlorzoxazone

Intra-assay variation of chlorzoxazone was determined by measuring three concentration levels (200, 100, 25 µM, or high, medium, low, respectively) three times in a single batch (*n* = 3). Mean activity was calculated from the following calibration curve linear equation: y = 0.0544 x + 0.0626 (r^2^ = 0.9997). The outcomes, summarized in Table 3, demonstrate that the relative standard deviation or % RSD (percentage of relative standard deviation) was <5% for chlorzoxazone. The experiment revealed that there was no large variation in the intra-assay experiment.

##### Intra-Assay Variation of 6-hydroxychlorzoxazone Metabolite

Determination of intra-assay variation of the metabolite was assessed by injecting three levels of 6-hydroxy chlorzoxazone concentrations (low, moderate, high) into a HPLC instrument three times in a single batch (*n* = 3). Mean activity was calculated from the following calibration curve linear equation: y = 0.0164 x − 0.0021 (r^2^ = 0.9994). The outcomes, summarized in Table 4, illustrate that the relative standard deviation or % RSD (percentage of relative standard deviation) was <5% for 6-hydroxychlorzoxazone. The experiment revealed that there is no large variation in the intra-assay experiment.

##### Inter-Assay Variation of Chlorzoxazone

Inter-assay variation was determined by measuring chlorzoxazone standards of three concentrations levels (200, 100, 25 µM, or high, medium, low, respectively) for three consecutive days, in separate batches. Mean activity was calculated from the following calibration curve linear equations on days 1, 2, and 3: y = 0.0544 x + 0.0626 (r^2^ = 0.9997) for day 1, y = 0.0522 x + 0.074 (r^2^ = 0.9994) for day 2, and y = 0.024 x − 0.0268 (r^2^ = 0.9994) for day 3. The results, summarized in Table 5, illustrate that the relative standard deviation or % RSD (percentage of relative standard deviation) was <10% for chlorzoxazone. The experiment revealed that there is no large variation between aliquots of the same batch sample in the inter-assay experiment.

##### Inter-Assay Variation of 6-Hydroxychlorzoxazone

Assessment of inter-assay variation of 6-hydroxychlorzoxazone was carried out by injecting three concentration levels of 6-hydroxychlorzoxazone (low, moderate, and high) for three consecutive days in separate batches (*n* = 3). Mean activity was calculated from the following calibration curve linear equations on days 1, 2, and 3: y = 0.0164 x − 0.0021 (r^2^ = 0.9994) for day 1, y = 0.0248 x − 0.0048 (r^2^ = 0.9999) for day 2, and y = 0.0277 x − 0.0141 (r^2^= 0.9996) for day 3. The outcomes are shown in Table 6, and illustrate that the relative standard deviation or % RSD was <10% for 6-hydroxychlorzoxazone. The experiment revealed that there is no variation between aliquots of the same batch sample in the inter-assay experiment.

#### 2.3.5. Stability Test

##### Stability Test for Chlorzoxazone

The stability of chlorzoxazone was investigated for three different concentrations (25, 100, and 200 µM) stored for 72 h at room temperature in natural light conditions. Phenacetin (used as an internal standard of 50 µM) was added to each batch. Each sample was analysed in triplicate (*n* = 3) for each batch. The calibration curve of chlorzoxazone was run at t = 0, t = 24, t = 48, and t = 72 h. The stability test results are summarized in the Table 7, below:

The outcomes in Table 7 revealed that there were no variations in the concentrations at 0, 24, 48, and 72 h compared to actual concentrations. Calibration curves were plotted for days 1–4 and all four calibration curves were as follows: day 1: y = 0.036 x − 0.0744 (r^2^ = 0.9998), day 2: y = 0.0356 x − 0.0103 (r^2^ = 0.9998), day 3: y = 0.0374 x − 0.132 (r^2^ = 0.9994), and day 4: y = 0.0358 x + 0.0298 (r^2^ = 0.9993), where the r^2^ met ICH guidelines. Percentage recovery values for chlorzoxazone at concentrations 25, 100, and 200 µM were found to be within acceptable criteria (80–120%) according to ICH guidelines. Thus, the results show high and acceptable accuracy (80–120%) for chlorzoxazone concentrations of 25, 100, and 200 µM, because the recovery was high (within acceptable range with ICH guidelines) compared with the standard known concentration. The results indicate that chlorzoxazone solution was stable for 72 h at ambient temperature, in accordance with the study done by Shaikh et al., (2008) [12].

##### Stability Test for 6-hydroxychlorzoxazone

Stability of 6-hydroxychlorzoxazone was investigated for three different concentrations (10, 40, and 80 µM) stored for 72 h at room temperature in natural light conditions. Phenacetin (used as an internal standard of 50 µM) was added to each batch. Each sample was analysed in triplicate (*n* = 3) for each batch. The calibration curve of 6-hydroxychlorzoxazone was run at t = 0, t = 24, t = 48, and t = 72 h. The stability test results are summarized in the Table 8, below.

The outcomes in Table 8 revealed that there were no variations in the concentrations at 0, 24, 48, and 72 h compared to actual concentrations. Calibration curves were plotted for days 1–4 and all four calibration curves were as follows: day 1: y = 0.0255 x − 0.0031 (r^2^ = 0.9999), day 2: y = 0.0259 x + 0.0141 (r^2^ = 0.9995), day 3: y = 0.0255 x − 0.0305 (r^2^ = 0.9991), and day 4: y = 0.0251 x – 0.0145 (r^2^ = 0.9993), where the r^2^ met within ICH guidelines. Percentage recovery values for 6-hydroxychlorzoxazone at concentrations 10, 40, and 80 µM were found to be within acceptable criteria (80–120%) according to ICH guidelines. Thus, the results shows high and acceptable accuracy (80–120%) for 6-hydroxychlorzoxazone concentrations of 10, 40, and 80 µM, because the recovery was high (within acceptable range with ICH guidelines) compared with the standard known concentration. The results indicate that 6-hydroxychlorzoxazone solution was stable for 72 h at room temperature, in accordance with the study done by Schelstraete et al. (2018) [13].

### 2.4. Effect of Salicylic acid on CYP2E1 Enzyme Activity

Different concentrations of chlorzoxazone (200, 150, 100, 50, and 25 µM) were incubated in the presence of 0, 10, 20, 40, and 60 µM of salicylic acid using 0.5 mg/mL of rat liver microsome with 1.0 mM nicotinamide adenine dinucleotide phosphate hydrogen (NADPH), 5.0 mM glucose-6-phosphate (G6P), 1.7 units/mL glucose-6-phosphate dehydrogenase (G6PDH), 1.0 mM ethylenediamine tetraacetic acid (EDTA), and 3.0 mM of magnesium chloride. The reaction was terminated at different sets of time. The incubation time was 40 min. The inhibitory effect of salicylic acid on CYP2E1 enzyme activity is shown in Figure 4 and Figure 5 below.

Studying CYP enzyme inhibition is considered as an important route in evaluating drug–drug interactions in the pharmaceutical field and in the drug development process. Several marketed drugs have been withdrawn from the market in the past few decades because CYP enzyme inhibition causes harmful drug–drug interactions [1]. To our knowledge, this systematic study seems to be the first time that the in vitro inhibitory effect of salicylic acid on the CYP2E1 probe substrate metabolism has been investigated.

The CYP2E1 isoform appears to be the most vulnerable when salicylic acid’s modulatory effect on each individual isoform is considered. Based on its characteristics, the CYP2E1 isoform has a broad substrate specificity, an ability to metabolise major marketed drugs, and is responsible for the catalysis of many carcinogens and poisons [2].

Only one enzyme exists in the CYP2E isoform and this is known as the CYP2E1 isozyme [14]. CYP2E1 has the ability to metabolize nitrosamines and short-chains such as ethanol, toluene, and paracetamol, and many other anaesthetics actively [15]. Therefore, the CYP2E1 enzyme can be included only in the metabolism of lower molecular weight drugs such as styrene, vinyl chloride, and benzene that are involved in the synthesis of many households cleaning products [16]. The substrates recognized by the CYP2E1 enzyme are chlorzoxazone, p-nitrophenol, coumarin, quinolone, and caffeine [17]. CYP2E1 can activate some of the above carcinogenic substances [14]. Bibi (2008) claimed that the expression of CYP2E1 might differ between genders. Likewise, CYP2E1 activity can be also affected by health conditions such as obesity and fasting, which may provide an apparent clarification for obesity associated with cancer [14].

In vitro evidence has also revealed that garlic and watercress serve as potent inhibitors of the CYP2E1 enzyme in the human body [18]. In addition, nervous system agents, such as isoflurane, ethanol, aflatoxin B1, balothane, and chlorzoxazone, can be preferably metabolised by the CYP2E1 enzyme [1].

Recent studies have suggested that the sensitivity of the CYP2E1 enzyme activity can be decreased by certain drugs and food stuffs for example green and black powder tea, ellagic acid, chrysin, N-acetyl cysteine, and dandelion [18]. Our previous study showed that salicylic acid acts as a non-competitive inhibitor for CYP2C11 enzyme activity [10]. Thus, salicylic acid has a low potency to cause toxicity and drug interaction with other drugs that are substrates for the CYP2C11 enzyme [10]. In this study (Figure 4; Figure 5), it is evident, based on the in vitro inhibition kinetic parameters in rat liver microsomes and Lineweaver plot shapes, that salicylic acid inhibited CYP2E1 activities with a mixed-type inhibition mode (competitive and non-competitive) with inhibition constant, K_i_ = 83.56 ± 2.730 µM and a 50% inhibitory concentration (IC_50_) value exceeding 100 µM (IC_5 0_= 167.12 ± 5.460 µM). In addition, salicylic acid at its concentration of 25 µM can considered as a saturated concentration, since its Michaelis constant K_m_ value represents the highest value compared to other salicylic acid concentrations. According to Table 9, K_m_ and the maximum rate of the reaction (V_max_) hanged between each concentration of salicylic acid, and thus salicylic acid acts as a mixed inhibition for CYP2E1 enzyme activity. Thus, in this study it is considered that the in vitro inhibitory effect of salicylic acid on CYP2E1 enzyme activity will be beneficial for a future in vivo study for healthcare screening of the effective use of salicylic acid in clinics and the safe administration of salicylic acid with other drugs, consistent with in vivo drug–drug interactions pharmacokinetic parameters. Some further investigational in vitro and in vivo studies in human CYP450s are needed due to the varied CYP2E isoform expression in species.

## 3. Materials and Methods

### 3.1. Chemicals and Reagents

6-Hydroxychlorzoxazone was purchased from Carbosynth limited (Compton, UK). Glucose-6-phosphate (G-6-P), glucose-6-phosphate dehydrogenase (G-6-PDH), ethylenediamine tetraacetic acid (EDTA), nicotinamide adenine dinucleotide phosphate (NADP^+^), magnesium chloride (MgCL_2_), and chlorzoxazone were purchased from Merck (Old Brickyard, Gillingham, UK). Methanol, acetonitrile, and water were of HPLC grade; salicylic acid, potassium phosphate monobasic, potassium phosphate dibasic, phenacetin with purity greater than 98%, and phosphoric acid (85% *w*/*w*) were purchased from Merck (Old Brickyard, Gillingham, UK).

### 3.2. Rat Liver Microsomes

Microsomes from liver, pooled from male rat (Sprague–Dawley) in this study were purchased from Merck (Old Brickyard, Gillingham, UK) and stored at –80 °C. The manufacturer had previously characterized the microsomes for CYP2E1 activity, CYP450 content, and protein concentrations.

### 3.3. Instruments

A Shimadzu LC-2010A HT high performance liquid chromatography system equipped with a low-pressure pump quaternary gradient (series 200 LC pump), a series 200 Peltier LC column oven for chromatographing the analysed solutions, a series 200 autosampler (Shimadzu, Tokyo, Japan), a degasser, and a model series 200 UV detector were used. The data were processed using the Shimadzu HPLC 2 Data Lab Solutions (Kingston University, UK) software processing system. A 5-µm C_18_ SUPLCO column (25 cm × 4.6 mm) was employed from Merck (Old Brickyard, Gillingham, UK) to separate target analytes using a low-pressure isocratic elution system. A UV-VIS Spectrometry instrument with a 1-cm-length quartz cuvette was purchased from VWR International Ltd. (Magna Park, Lutterworth, Leicestershire, LE17 4XN, UK) and UV spectra were processed using a Bio 100 Cary software from Agilent Technologies LDA (Cheadle Royal Business Park, Stockport, Cheshire, SK8 3GR, UK). A 570 pH meter, purchased from JENWAY Limited (Beacon Road, Stone, Staffordshire, ST15 0SA, UK) was used.

### 3.4. CYP450 Assay

#### 3.4.1. CYP2E1 Substrate and its Metabolite

Validations were conducted using HPLC system LC-2010A HT Module (Shimadzu, Tokyo, Japan). Chromatographic experiments were evaluated in a low-pressure isocratic system. The separations of the four target components (salicylic acid as a tested inhibitor, phenacetin as an internal standard, chlorzoxazone as CYP2E1 substrate, and 6-hydroxychlorzoxazone as CYP2E1 metabolite) were injected into a SUPELCO C18 column (25 cm × 4.6 mm, 5 µm particle size). The mobile phase for the chromatographic separation of the four compounds was made up as follows: (A): 60% H_2_O, (B): 15% of methanol, and (C): 25% of acetonitrile. The flow rate was set to 0.7 mL/min, and the oven temperature was set at 25 °C. Wavelength detection was set to 282 nm. An injection volume of 10 µL was used. The mobile phase consisted of methanol: water: acetonitrile (15:60:25 % *v*/*v*). This provided a good separation and resolution of CYP2E1 components.

#### 3.4.2. Inhibition of CYP2E1 Enzymatic Activity Assay

CYP2E1 activity was assessed using chlorzoxazone (substrate) via the formation of 6-hydroxychlorzoxazone (the metabolite for the CYP2E1 enzyme). Oxidative metabolism of chlorzoxazone was evaluated using a NADPH-regenerating system consisting of: 1.0 mM nicotinamide adenine dinucleotide phosphate hydrogen (NADPH), 5 mM glucose-6-phosphate (G6P), 1.7 units/mL glucose-6-phosphate dehydrogenase (G6PDH), 1.0 mM ethylenediamine tetraacetic acid (EDTA), and 3.0 mM magnesium chloride. A 500-µL incubation volume containing NADPH-regenerating system and a final concentration of 0.067 M potassium phosphate buffer at pH = 7.4 was incubated in triplicate. The incubation mixture consisted of 0.5 mg/mL of pooled liver microsomes with a serial range of chlorzoxazone concentrations (25, 50, 100, 150, and 200 µM, dissolved in mobile phase) and a serial range of salicylic acid concentrations (0, 10, 25, 40, and 60 µM, dissolved in mobile phase). Pre-incubation of all components was done for 10 min in a water bath at T = 37 °C before the addition of nicotinamide adenine dinucleotide phosphate (NADP^+^) into the mixture [19]. The final concentration of organic solvent did not exceed the 1% *v*/*v*. Tubes were incubated for 40 min in an Eppendorf thermomixer (Eppendorf UK Limited, Stevenage) at 150 rpm (37 °C). Termination of the reaction was carried out by adding ice-cold acetonitrile containing 50 µM of phenacetin as an internal standard at each time interval (10, 15, 20, 25, 30, 35, and 40 min). Tubes were centrifuged in a microcentrifuge (13,000× *g*) for 12 min to precipitate protein. Then the supernatant was collected and dissolved in a mobile phase (60% H_2_O, 15% methanol, and 25% acetonitrile) made up to 1000 µL volume. A 10-µL injection volume of the dissolved supernatant was used for HPLC analysis. 

### 3.5. Selection of Analytical Wavelength:

#### CYP2E1 Assay

UV-VIS spectrometry using methanol as a blank was evaluated for phenacetin (50 µM), salicylic acid (100 µM), chlorzoxazone (200 µM), and 6-hydroxychlorzoxazone (50 µM) standard solutions in the UV region of 200–350 nm.

### 3.6. Preparation of Mobile Phase:

#### CYP2E1 Assay

Various mobile phases for CYP2E1 assay were tested. The most suitable mobile phase was: HPLC grade methanol (low UV cut-off of 205 nm) as mobile phase (A), H_2_O as mobile phase (B), and HPLC grade acetonitrile as mobile phase (C) (A: 15%, B: 60%, C:25%).

### 3.7. Preparation of Standard and Sample Solutions

#### Analytes and Metabolite Standard Solution Preparation

Salicylic acid (SA) (1.38 mg) (C = 200 µM) was measured accurately and dissolved in a 50-mL volumetric flask by a mobile phase (60% H_2_O, 25% acetonitrile, and 15% methanol). Serial dilutions were performed, yielding final concentrations of 40, 25, and 10 µM. Chlorzoxazone (3.39 mg) (C = 400 µM) was weighed accurately and dissolved in a mobile phase (60% H_2_O, 25% acetonitrile, and 15% methanol). A serial dilution of chlorzoxazone stock solution was made, yielding to final concentrations of 300, 200, 150, 100, 50, and 25 µM. Phenacetin powder (0.9 mg) was weighed and dissolved in pure HPLC-grade acetonitrile. The metabolite for the CYP2E1 enzyme (6-hydroxychlorzoxazone) was prepared as a stock solution of C = 100 µM, followed by a range of serial dilutions (80, 60, 40, 20, and 10 µM).

### 3.8. Data Analysis:

Standard/calibration curves consisted of different concentration ranges of chlorzoxazone, 6-hydroxychlorzoxane, and a fixed concentration of phenacetin (50 µM) as an internal standard. Validation parameters (Range, LOD, LOQ, % error, recovery, and accuracy) were calculated using Microsoft Excel 2010 software. All results are presented as mean ± S.D.

Formation of the 6-hydroxychlorzoxazone metabolite of the tested CYP2E1 substrate (chlorzoxazone) was measured for CYP inhibition analysis. Microsoft Excel 2010 software was used for determination of area peak ratios of both metabolite and internal standard. Pharmacokinetic parameters (*V*_m_, *K*_m_, *Cl*_int_ (hepatic intrinsic clearance), *K*_i_) were determined by nonlinear regression analysis from secondary Lineweaver–Burk and Michaelis–Menten plots of the enzyme activity-metabolite concentration data. The type of CYP2E1 inhibition was assumed to be mixed inhibition based on the shape of Lineweaver–Burk plots, the standard error, the Akaike information criterion (AIC), and Schwarz criterion (SC) values, which were fitted by a simple non-linear regression model. A non-linear regression model was chosen as a better fit because of its low AIC value. The Akaike weights (∆AIC) are the relative difference between the best model and each other model in the set. The ∆AIC provided substantial evidence for the model (∆AIC < 2). The 50% inhibitory concentration (IC_50_) value was calculated by using the following mixed inhibition equation:V= [V_0_/(1+(I/IC_50_)^S^)](1)
where V is the observed velocity, V_0_ is uninhibited velocity, S is slope factor, and I is the inhibitor concentration.

## 4. Conclusions

In conclusion, all the analytical parameters (accuracy, precision, % error, % recovery, LOD, LOQ, linear regression) for chlorzoxazone and 6-hydroxychlorzoxazone were in line with ICH guidelines. The in vitro data provided allow us to understand the drug interactions with salicylic acid. Nevertheless, our findings indicated that salicylic acid potentially acts as a mixed inhibitor (competitive and non-competitive inhibitor) for CYP2E1 enzyme activity in rat liver microsomes. This finding provides useful data for the efficacy and the safe use of salicylic acid in clinical practice. However, the extent to which our obtained data can be related to human CYP isoforms is unknown. The majority of clinically important drugs such as S-warfarin, steroids, and retinoic acid are metabolised by both rat CYP2C11 and human CYP2C9 that display a high identity of amino acid sequences. Thus, it is primarily important to investigate whether human CYP isoforms are induced by salicylic acid, and further verification of the obtained in vitro data should be done with in vivo experiments.

## Figures and Tables

**Figure 1 molecules-25-00932-f001:**
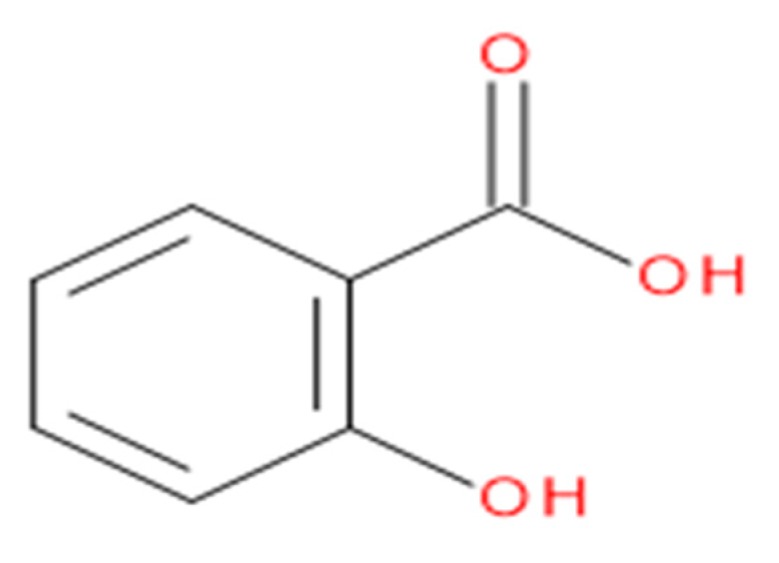
Chemical structure of salicylic acid.

**Figure 2 molecules-25-00932-f002:**
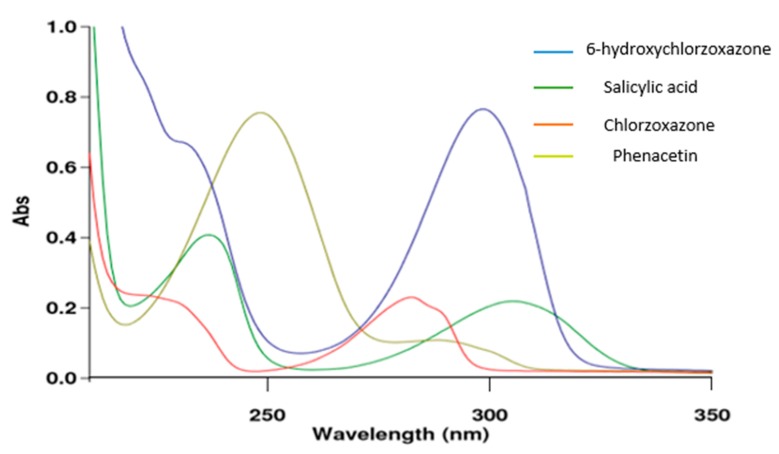
Overlain spectra of each component in the cytochrome P2E1 (CYP2E1) assay.

**Figure 3 molecules-25-00932-f003:**
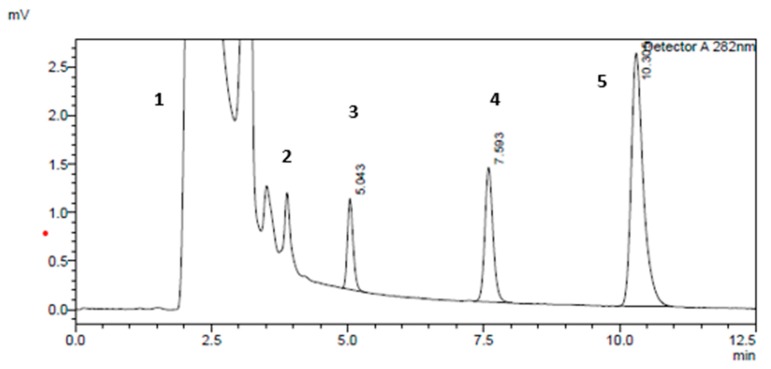
Typical HPLC chromatogram of CYP2E1 components with standard rat microsomal medium (40 min incubation of chlorzoxazone) at 282 nm wavelength detection with a 200-µM chlorzoxazone concentration. The peaks marked are: (**1**) the nicotinamide adenine dinucleotide phosphate hydrogen (NADPH)-regenerating system, (**2**) salicylic acid, (**3**) 6-hydroxychlorzoxazone, (**4**) phenacetin, and (**5**) chlorzoxazone, respectively.

**Figure 4 molecules-25-00932-f004:**
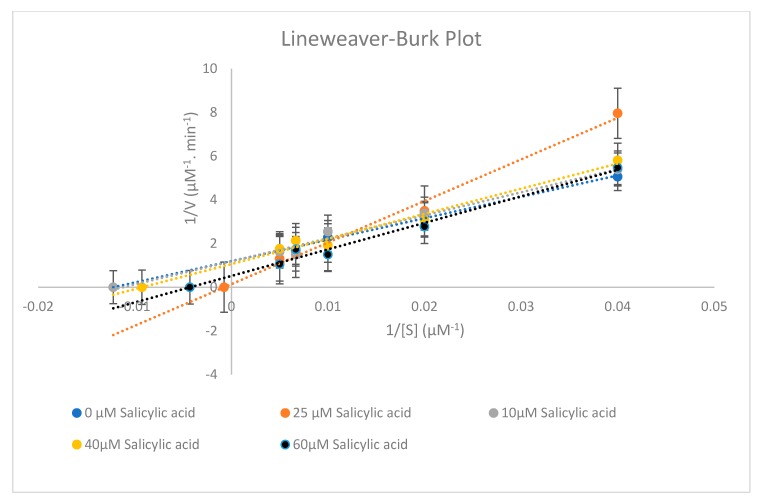
Representative Lineweaver–Burk plot for the inhibition of CYP2E1-catalysed chlorzoxazone 6-hydroxylation (chlorzoxazone (25–200 µM)) with 0, 10, 25, 40, and 60 µM salicylic acid. Each point represents the average of three determinations.

**Figure 5 molecules-25-00932-f005:**
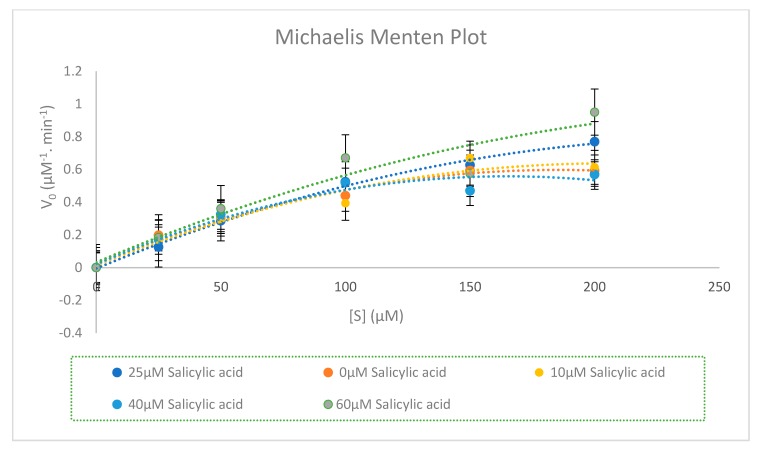
Representative Michaelis–Menten plot for the inhibition of CYP2E1-catalysed chlorzoxazone 6-hydroxylation (chlorzoxazone (25, 50, 100, 150, and 200 µM)) by 0, 10, 25, 40, and 60 µM salicylic acid. The curves were fitted with a polynomial function of order 2. Each point represents the average of three determinations.

**Table 1 molecules-25-00932-t001:** Analytical performance.

Standards	Chlorzoxazone	6-Hydroxychlorzoxazone
Regression equation	y = 0.0544 x + 0.0626	y = 0.0164 x – 0.0021
*r^2^*	0.9997	0.9994
Linear range	25–400 µM	10–100 µM

**Table 2 molecules-25-00932-t002:** LOD and LOQ for chlorzoxazone and 6-hydroxychlorzoxazone.

Standards	Chlorzoxazone	6-Hydroxychlorzoxazone
Limit of Detection (LOD)	4.972 µM	1.997 µM
Limit of Quantitation (LOQ)	15.068 µM	6.052 µM

**Table 3 molecules-25-00932-t003:** Intra-assay variation for chlorzoxazone (*n* = 3).

Chlorzoxazone Standard	Mean Activity (µM)	Standard Deviation	Relative Standard Deviation (%)
Low activity standard (C = 25 µM)	36.424	1.014	2.785
Medium activity standard (C = 100 µM)	106.421	0.727	0.684
High activity standard (C = 200 µM)	218.814	0.599	0.274

% RSD: Percentage of relative standard deviation.

**Table 4 molecules-25-00932-t004:** Intra-assay variation of CYP2E1 enzyme metabolite (6-hydroxychlorzoxazone) (*n* = 3).

6-Hydroxy Chlorzoxazone Standard	Mean Activity (µM)	Standard Deviation	Relative Standard Deviation (%)
Low activity standard (C = 10 µM)	10.114	0.038	0.385
Moderate activity standard (C = 40 µM)	41.058	0.291	0.709
High activity standard (C = 80 µM)	81.299	0.101	0.125

% RSD: Percentage of relative standard deviation.

**Table 5 molecules-25-00932-t005:** Inter-assay variation for chlorzoxazone.

Chlorzoxazone Standard (µM)		Mean Area Peak (*n* = 3 each level)	Mean^a^ Activity (µM)	Standard Deviation (STD)	Relative Standard Deviation (RSD (%))
Low activity standard (C = 25 µM)	Day 1	1.489	25.245	0.696	2.756
	Day 2	1.373			
	Day 3	0.564			
Medium activity standard (C = 100 µM)	Day 1	5.668	105.088	7.583	7.216
	Day 2	6.089			
	Day 3	2.297			
High activity standard (C = 200 µM)	Day 1	11.719	202.275	8.525	4.214
	Day 2	10.378			
	Day 3	4.657			

% RSD: Percentage of relative standard deviation. ^a^ Mean concentration (µM).

**Table 6 molecules-25-00932-t006:** Inter-assay performance of 6-hydroxychlorzoxazone.

6-Hydroxy Chlorzoxazone Standard (µM)		Mean Area Peak (*n* = 3 each level)	Mean Activity (µM)	Standard Deviation (STD)	Relative Standard Deviation (RSD(%))
Low activity standard (C = 10 µM)	Day 1	0.161	9.955	0.037	0.367
	Day 2	0.242			
	Day 3	0.263			
Medium activity standard (C = 40 µM)	Day 1	0.661	41.335	2.219	5.370
	Day 2	1.096			
	Day 3	1.072			
High activity standard (C = 80 µM)	Day 1	1.329	80.498	0.783	0.973
	Day 2	2.002			
	Day 3	2.185			

% RSD: Percentage of relative standard deviation.

**Table 7 molecules-25-00932-t007:** Stability test data of chlorzoxazone.

Analytical Parameters	Nominal Level (Actual Concentration of Chlorzoxazone (µM))			
		25	100	200
**Calculated concentration (µM)**	0 h	26.635	99.418	197.857
	24 h	24.359	96.815	195.943
	48 h	26.942	96.025	190.029
	72 h	23.433	97.834	195.523
**% Recovery ^a^**	24 h	91.456	97.382	99.033
	48 h	101.151	96.586	96.044
	72 h	88.246	98.407	98.821
**Accuracy ^b^ (%)**	0 h	93.459	100.582	101.072
	24 h	102.562	103.185	102.029
	48 h	92.232	103.975	104.985
	72 h	106.267	102.166	102.239

^a^ % Recovery= (concentration of chlorzoxazone at 24 h)/ Standard concentration of chlorzoxazone) × 100. ^b^ Accuracy = ((calculated concentration - actual concentration)/actual concentration)×100.

**Table 8 molecules-25-00932-t008:** Stability test data of 6-hydroxychlorzoxazone.

Analytical Parameters	Nominal Level (Actual Concentration of 6-Hydroxychlorzoxazone (µM))			
		10	40	80
**Calculated concentration (µM)**	0 h	9.537	40.431	81.391
	24 h	8.848	38.719	77.154
	48 h	9.039	41.731	80.298
	72 h	10.580	42.359	83.591
**% Recovery ^a^**	24 h	92.774	95.767	94.794
	48 h	94.782	103.217	98.657
	72 h	110.099	104.772	102.702
**Accuracy ^b^ (%)**	0 h	104.631	98.923	98.261
	24 h	111.522	103.202	103.558
	48 h	109.608	95.672	99.627
	72 h	94.200	94.100	95.512

^a^ % recovery = (concentration of 6-hydroxychlorzoxazone at 24 h)/ standard concentration of 6-hydroxychlorzoxazone) × 100. ^b^ Accuracy = 100 − ((calculated concentration − actual concentration)/actual concentration) × 100.

**Table 9 molecules-25-00932-t009:** Calculated parameters of enzyme metabolism for the in vitro CYP2E1 inhibition study.

Parameters of Enzyme Metabolism	0 µM Salicylic Acid	10 µM Salicylic Acid	25 µM Salicylic Acid	40 µM Salicylic Acid	60 µM Salicylic Acid
K_m_ (µM)	88.731 ± 6.003	81.732 ± 6.517	1322.751 ± 0.403	107.991 ± 4.932	232.306 ± 0.762
V_max_ (µM^−1^.min^−1^)	0.842 ± 0.186	0.834 ± 0.220	6.954 ± 0.033	0.939 ± 0.154	1.915 ± 0.972
Cl_int_ (µM^−2^.min^−1^)	0.0094 ± 0.202	0.0102 ± 0.186	0.0053 ± 0.359	0.0087 ± 0.218	0.0082 ± 0.002

K_m_: Michaelis constant. V_max_: Maximum rate of the reaction. Cl_int_: Hepatic intrinsic clearance.

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
