# Peer review of "Validation of an HPLC Method for the Simultaneous Quantification of Metabolic Reaction Products Catalysed by CYP2E1 Enzyme Activity: Inhibitory Effect of Cytochrome P450 Enzyme CYP2E1 by Salicylic Acid in Rat Liver Microsomes"

_molecules, 2020, doi:10.3390/molecules25040932_

Round 1
Reviewer 1 Report
The focus of the study was to determine the inhibitory effect of salicylic acid on CYP2E1 activity on rat liver microsomes. The mass spectrometry work was fairly well done. However, the major issue of their paper is that salicylic acid has mixed state inhibition based on one concentration of salicylic acid (25 mM). If you look at the error bars in Figure 5, there is no real statistical difference between the curves. I think the authors need more data to demonstrate the inhibitory mechanism. Without the 25 µM salicylic acid data point, I am not convinced that salicylic acid inhibits chlorzoxazone 6-hydroxylation much. I think the authors need to use more concentrations of chlorzoxazone and salicylic acid to conclusively demonstrate that salicylic acid inhibits chlorzoxazone 6-hydroxylation by CYP2E1. I am leaning toward rejection, but if the authors can provide appropriate data to conclusively confirm mixed-state inhibition, I think that the manuscript would be acceptable for publication in the journal Molecules.
The authors should explain why they used phenacetin as an internal standard and why it is preferable to other internal standards. Table 9 title is a little confusing because it implies that pharmacokinetics was performed. I would suggest that the authors change the title to: Pharmacokinetic parameters calculated from in vitro measurements in our CYP2E1 inhibition study. Abstract: I would not consider a Km of 84 µM and an IC50 of 167 mM as being potent. I would recommend that the authors eliminate the word potent. I would not consider those numbers as potent. The enzyme kinetics in Figure 4, Figure 5 and Table 9 are confusing. The Vmax and Km values at 25 µM salicylic acid seems like an outlier. In particular, the error bars in for 25 mM salicylic acid have a lot of overlap with other values of salicylic acid. I think that it would be very helpful for the authors to perform this experiment at additional salicyclic concentrations to show that the effects of 25 µM salicylic acid are not an outlier. The authors do not provide standard deviations in Table 9. They should add that to the table. The authors should explain how Figure 5 was fit. One of the curves look like it was fit with a parabolic function.Author Response
Firstly, we would like to thank the reviewer for reviewing our paper.
Point-by-point response (reviewer's comments have been paraphrased):
"The authors need more data to demonstrate the inhibitory mechanism".
1. As requested, an additional experiment has now been conducted using 60 μM of salicylic acid. This confirmed that salicylic acid acts as a mixed inhibitor for CYP2E1 enzyme (see Figure 4 and Table 9 of the manuscript file below). Figure 5 now showed a real statistical difference between the curves, when this additional experiment was carried out (60 μM of salicylic acid inhibition study).
"Why has phenacetin been used as an internal standard?"
2. Several different internal standards were used during method development such as: carbamazepine, aspirin, caffeine and paracetamol. However, all these compounds co-eluted with 6-hydroxychlorzoxazone metabolite peak when using the same mobile phase (60% H2O,15% acetonitrile, 25% methanol). The most suitable internal standard found was phenacetin, since we obtained the best resolution. In addition to this, phenacetin has been recommended for use as an internal standard in CYP2E1 assay of many inhibition studies (Min Sun, et al., 2014, "Inhibitory effect of celastrol on rat liver cytochrome P450 1A2, 2C11, 2D6, 2E1 and 3A2 activity". Moreover, phenacetin does not interact with CYP2E1 substrate (chlorzoxazone) and with the tested inhibitor (salicylic acid).
"The authors change the title of Table 9".
3. Table 9 title has been changed to “Calculated parameters of enzyme metabolism for the in vitro CYP2E1 inhibition study”.
"The word "potent" be removed from the abstract".
4. The word “Potent” been omitted from the abstract and conclusions section.
"Perform experiment at an additional salicylic acid concentration to verify findings".
5. An additional 60 μM salicylic acid inhibition study was carried out and the results confirmed that salicylic acid acts as a mixed inhibitor for CYP2E1 enzyme (different Km and Vm from 0 μM salicylic acid).
"Provide standard deviations to Table 9."
6. Standard deviation have now been added to Table 9.
"How were curves in Fig. 5 fit?"
7. Figure 5 curves were fitted with a polynomial function of order 2. This information has been added to the figure title.

Reviewer 2 Report
This paper is a more or less a mirror image of a preceeding one published by the same authors dealing with inhibition of another CYp enzyme, 2C11 (Molecules 2019, 24, 4294, doi:10.3390/molecules24234294) by salicylate. Interestingly, in the case of CYP2C11, inhibition with Ki=84.5 microM is presented as weak, and this with CYP2E1, Ki=83.5 microM as a potent one. The paper has been written with little care and needs a detailed check of the style and also typos (e.g. lines 33, 34 - "only the CYP1,CYP2 and CYP3 families - however with more than 10 forms involved, thorough the whole paper, upper case/lower case letters mixed as Testosterone or testosterone, Phosphate or phosphate, Chlorzoxazone or chlorzoxazone ....etc., Fig. 4 - Lineweaver-Burk plot but in the legend it is the Dixon one (?), Table 9 - Pharmacokinetic parameters? rather parameters of enzyme metabolism - PKinetics needs AUC, k, t1/2 etc - there is no pharmacokinetics shown, CYP450 is a wrong name, correct ones are CYP or P450, line 356 - volumetric flask mobile phase..? , line 266: hepatic liver?..). Also the literature data are misleading - e.g. regarding the expression of CYP2E1 and substrates (see e.g. F.P. Guengerich in P.R. Ortiz de Montellano (Ed.) book on Cytochrome P450 (4th Edn., Springer 2015)), refs. should be also checked, e.g. Ref. 13 (Bibi Z.) was retracted, data presentation: Akaike and Schwarz criteria - not described, which candidate models chosen, how was the model fitting done, what were Akaike weights?, Ki=IC50/2 cannot be used, it is valid for competitive inhibition, here, a mixed inhibition was found (!).
Conceptual comment: the rather good analytical work should rather be done with human liver microsomes - this would be muc more useful.
Human microsomes not rat
Author Response
Firstly, we would like to thanks the reviewer for the great job in reviewing our paper.
Point-by-point response:
The paper has been reviewed by two native speakers for English language editing.
1) Line 33-34: (Only the CYP1, CYP2 and CYP3 families however with more than 10 forms involved) has been changed, as recommended.
2) The name of the substances were changed and corrected to lower case letters.
3) Figure 4: Word “Dixon” is omitted and has been replaced by Lineweaver-Burk plot.
4)Table 9: Pharmacokinetic parameters been replaced by “Calculated parameters of enzyme metabolism for the in vitro CYP2E1 inhibition study”.
5) “CYP450” been changed to P450 and CYPs, as requested.
6) Line 356 has been corrected (see Line 565-566 of the manuscript file).
7) Line 266: “hepatic liver been replaced by “rat liver microsomes “ (see line 403 of the manuscript file).
8) "literature data are misleading"- the sentence regarding the expression of CYP2E1 and substances been deleted and replaced by another sentence (see Line 392-393 of the manuscript file).
9) Reference 13 been corrected, as requested (see Reference 13 in the reference section of the manuscript file).
10) Akaike and Schwarz criteria have been described and a non-linear regression model was chosen as a better fit. Akaike weights were <2, which is substantial evidence for the model (see Lines 585-589 of the manuscript file).
11) Ki=IC50/2 equation been replaced by a different equation (see Line 590 of the manuscript file).
12) Yes, point taken; the P450 content protein in rats are 673 pmol/mg, which is greater than P450 content protein in humans (307 pmol/mg) (reference M. Pasanen 2004 “Species differences in CYP enzymes"). However, many studies use rat microsomes and this still gives useful data.

Reviewer 3 Report
In this study, after setting the conditions for accurately measure in rat microsomes, the activity of CYP2E1 by determining the substrate (chlorzoxazone) and the resulting metabolite (6-hydroxy-chlorzoxazone) using HPLC, the authors have carried kinetic analysis in the presence and the absence of salicylic acid.
Main points
1) The use of significant digits is wrong throughout the manuscript. The authors should revise the number of positions that are appropriate for the fractional parts according to the magnitude of the integral ones following the error theory.
2) The Introduction section is too long and not focused on the actual content of the experimental work and the title. For instance, to define fundamental concepts, such as pharmacokinetics and pharmacodynamics, is not necessary.
3) Figure 5. Considering the high SD values and the small differences among mean values, the credibility of the conclusion derived from this figure is unclear to me.
4) Although these studies were carried out in rat microsomes, which means that extrapolation must be done with caution, an important question that should be clearly stated in the Discussion section is how relevant kinetic parameters of salicylic acid-induced CYP2E1 inhibition are in the clinical context. Are Ki values consistent with drug-drug interaction in vivo?
5) The whole manuscript requires improvement regarding consistency. The name of all compounds should be lower case (inconsistently, some are capitalized). In tables and figure legends, “%” must be used consistently. Between values and units must always be a space.
Author Response
Firstly, we would like to thank the reviewer for the great job in reviewing our paper.
Point-by-point response:
1) The use of significant digits been corrected, as recommended and each value was rounded to 3 decimal numbers (See the manuscript file).
2) Pharmacokinetics and Pharmacodynamics definitions were removed from the introduction part as recommended. (See the introduction section in the manuscript file).
3) Lineweaver-Burk plot (1/v vers 1/[S]) is derived from Michaelis-Menten plot (V versus [S]). Each concentration of salicylic acid acid was fitted as a polynomial function of order 2. The aim of presenting figure 5 in the manuscript is to show that at different concentration of salicylic acid Vm is different (Vm of each concentration of salicylic acid is the maximum V point in Michaelis-Menten plot). Additionally, the aim of presenting figure 5 also to show how Lineweaver-Burk plot was obtained. In our opinion, the precision we have obtained in these biochemical experiments is in keeping with other co-workers and does not detract from the significance of the results obtained.
4) This question been answered in the discussion part (See the manuscript file) (lines 409-412 and 607-612).
5) Names of all compounds been revised and been replaced by a lower case letter. % been used consistently in tables and figures legends. A space has been used between units and values, as recommended (see the manuscript file).

Round 2
Reviewer 1 Report
They have done a pretty good job addressing my concerns.
Reviewer 2 Report
Can be published in this form. Selection of papers for Introduction is peculiar, papers from respected authors would add to the feeling that the authors are oriented in this field.